# Replication Study: Systematic identification of genomic markers of drug sensitivity in cancer cells

**John P Vanden Heuvel[1,2], Ewa Maddox[1], Samar W Maalouf[1], Reproducibility Project: Cancer Biology***

[1]Indigo Biosciences, State College, United States; [2]Department of Veterinary and Biomedical Sciences, Pennsylvania State University, State College, United States

***For correspondence:**
nicole@scienceexchange.com;
tim@cos.io

**Group author details:**
Reproducibility Project: Cancer
Biology See page 11

**Competing interest:** See
page 11

**Reviewing editor:** Joaquín M
Espinosa, University of Colorado
School of Medicine, United
States

**Abstract** In 2016, as part of the Reproducibility Project: Cancer Biology, we published a Registered Report (Vanden Heuvel et al., 2016), that described how we intended to replicate selected experiments from the paper 'Systematic identification of genomic markers of drug sensitivity in cancer cells' (Garnett et al., 2012). Here we report the results. We found Ewing's sarcoma cell lines, overall, were more sensitive to the PARP inhibitor olaparib than osteosarcoma cell lines; however, while the effect was in the same direction as the original study (Figure 4C; Garnett et al., 2012), it was not statistically significant. Further, mouse mesenchymal cells transformed with either the *EWS-FLI1* or *FUS-CHOP* rearrangement displayed similar sensitivities to olaparib, whereas the Ewing's sarcoma cell line SK-N-MC had increased olaparib sensitivity. In the original study, mouse mesenchymal cells transformed with the *EWS-FLI1* rearrangement and SK-N-MC cells were found to have similar sensitivities to olaparib, whereas mesenchymal cells transformed with the *FUS-CHOP* rearrangement displayed a reduced sensitivity to olaparib (Figure 4E; Garnett et al., 2012). We also studied another Ewing's sarcoma cell line, A673: A673 cells depleted of *EWS-FLI1* or a negative control both displayed similar sensitivities to olaparib, whereas the original study reported a decreased sensitivity to olaparib when *EWS-FLI1* was depleted (Figure 4F; Garnett et al., 2012). Differences between the original study and this replication attempt, such as the use of different sarcoma cell lines and level of knockdown efficiency, are factors that might have influenced the outcomes. Finally, where possible, we report meta-analyses for each result.
DOI: https://doi.org/10.7554/eLife.29747.001

## Introduction

The Reproducibility Project: Cancer Biology (RP:CB) is a collaboration between the Center for Open Science and Science Exchange that seeks to address concerns about reproducibility in scientific research by conducting replications of selected experiments from a number of high-profile papers in the field of cancer biology (*Errington et al., 2014*). For each of these papers a Registered Report detailing the proposed experimental designs and protocols for the replications was peer reviewed and published prior to data collection. The present paper is a Replication Study that reports the results of the replication experiments detailed in the Registered Report (*Vanden Heuvel et al., 2016*) for a 2012 paper by Garnett et al., and uses a number of approaches to compare the outcomes of the original experiments and the replications.

In 2012, Garnett et al. reported the results of a large-scale high throughput screen to identify novel interactions between investigational drugs and cancer-derived human cell lines, along with a similar study published at the same time by Barretina and colleagues (*Barretina et al., 2012*). In addition to capturing expected gene-drug interactions, several unpredicted associations were

identified, including an enhanced sensitivity between the *EWS-FLI1* translocation of Ewing's sarcoma family tumors and poly(ADP-ribose) polymerase (PARP) inhibitors (*Garnett et al., 2012*). Selective inhibition of cell survival and proliferation in Ewing's sarcoma cell lines was observed with the PARP inhibitor, olaparib, comparable to the observed inhibition in *BRCA*-deficient cells (*Garnett et al., 2012*). Further, the *EWS-FLI1* translocation was reported to be sufficient for increased sensitivity of cells to olaparib, while transient depletion of *EWS-FLI1* from Ewing's sarcoma cells resulted in partial rescue of olaparib sensitivity, suggesting the sensitivity of Ewing's sarcoma cells to olaparib might be related to EWS-FLI1 transcriptional activity.

The Registered Report for the 2012 paper by Garnett et al. described the experiments to be replicated (Figure 4C and E–F, and Supplemental Figures 16 and 20), and summarized the current evidence for these findings (*Vanden Heuvel et al., 2016*). Additional studies have reported hypersensitivity of Ewing's sarcoma cell lines to PARP inhibitors (*Brenner et al., 2012*; *Engert et al., 2015*; *Gill et al., 2015*; *Norris et al., 2014*; *Ordóñez et al., 2015*; *Smith et al., 2015a*; *Stewart et al., 2014*). However, studies extending the use of olaparib, or other PARP inhibitors, as monotherapies in xenograft models have reported limited effectiveness (*Norris et al., 2014*; *Ordóñez et al., 2015*; *Smith et al., 2015a*; *2015b*; *Stewart et al., 2014*), consistent with no objective responses from a phase II study of olaparib (*Choy et al., 2014*). In agreement with these observations, a new methodology for biomarker discovery, that accounts for variability in general levels of drug sensitivity, failed to find a statistically significant association of PARP inhibitors and the *EWS-FLI1* translocation (*Geeleher et al., 2016*). However, studies testing combinatorial treatments of PARP inhibitors with other drugs, such as the DNA alkylating agent temozolomide, have reported enhanced sensitivity of Ewing sarcomas (*Brenner et al., 2012*; *Engert et al., 2015*; *Gill et al., 2015*; *Norris et al., 2014*; *Ordóñez et al., 2015*; *Smith et al., 2015b*; *Stewart et al., 2014*), with several clinical trials beginning (*Pishas and Lessnick, 2016*). Furthermore, a recent study reported that cells with *SLFN11* inactivation are more resistant to PARP inhibitors, as single agents or in combination with temozolomide; however combination with an ATR inhibitor can overcome this resistance (*Murai et al., 2016*).

The outcome measures reported in this Replication Study will be aggregated with those from the other Replication Studies to create a dataset that will be examined to provide evidence about reproducibility of cancer biology research, and to identify factors that influence reproducibility more generally.

## Results and discussion

### Sensitivity of Ewing's sarcoma cell lines to PARP inhibition

We sought to independently replicate whether Ewing's sarcoma cell lines were more sensitive to the PARP inhibitor, olaparib, than control cell lines. This experiment is comparable to what was reported in Figure 4C and Supplemental Figure 16 of *Garnett et al. (2012)* and described in Protocol 1 in the Registered Report (*Vanden Heuvel et al., 2016*). While the original study included a comparison of Ewing's sarcoma cells to cell lines from other tumor types, this replication attempt was restricted to osteosarcoma cells. Similar to the original study, a *BRCA2*-deficient cell line (DoTc2-4510) and a *BRCA*-proficient cell line (MES-SA) were included to capture differential sensitivity to olaparib across genotypes. Olaparib sensitivity was determined for each cell line using a colony formation assay over a range of concentrations (0.1, 0.32, 1, 3.2, or 10 µM) with the effective concentration determined as the olaparib concentration at which the number of colonies were reduced by at least 90% compared to vehicle control. Similar to the original study, DoTc2-4510 cells were found to be highly sensitive to olaparib (0.1 µM), while MES-SA were largely resistant (10 µM) (*Figure 1*). The median effective concentration of osteosarcoma cell lines (n = 7) was 3.2 µM (range: 3.2 µM - 10 µM), while the median effective concentration of Ewing's sarcoma cell lines (n = 5) was 1 µM (range: 0.1 µM - 10 µM) (*Figure 1*). This compares to the original study that reported a median effective concentration of 3.2 µM (range: 1 µM - 10 µM) for osteosarcoma cell lines (n = 8) and 1 µM (range: 0.32 µM - 1 µM) for Ewing's sarcoma cell lines (n = 5) (*Garnett et al., 2012*).

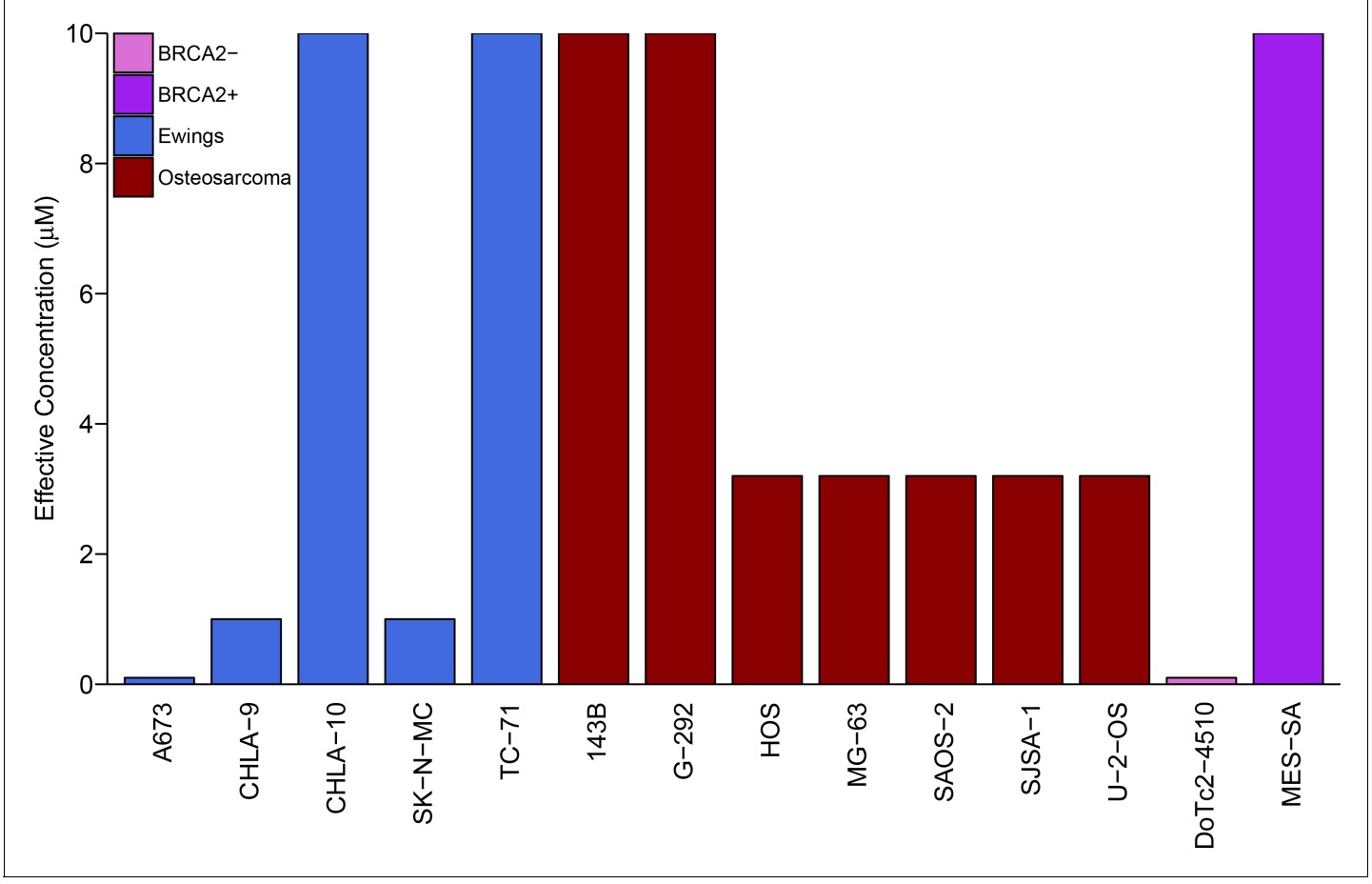

**Figure 1.** Sensitivity of Ewing's sarcoma cell lines to olaparib. Colony formation assays were performed on the indicated cell lines in the presence of a range of olaparib concentrations (0.1, 0.32, 1, 3.2, or 10 µM) or vehicle control (DMSO). Plates were retreated every 3 or 4 days, fixed and stained 7–21 days following plating, and colonies counted. The effective concentration displayed for each cell line is defined as the concentration that reduced colony formation by greater than 90% compared to vehicle control. For G-292 cells, where the highest olaparib concentration tested (10 µM) did not inhibit colony formation by at least 90%, the effective concentration was defined as 10 µM. Wilcoxon-Mann-Whitney test for ordinal data comparing the effective concentrations of Ewing's sarcoma cell lines to osteosarcoma cell lines; $U$ = 12, $p$=0.390. Additional details for this experiment can be found at https://osf.io/zy3s5/.
DOI: https://doi.org/10.7554/eLife.29747.002

The following figure supplement is available for figure 1:

**Figure supplement 1.** Repeat of colony formation assay.
DOI: https://doi.org/10.7554/eLife.29747.003

To compare the sensitivity of Ewing's sarcoma cells to osteosarcoma cells, a Wilcoxon-Mann-Whitney test for ordinal data was performed using the effective concentration as outlined in the Registered Report (*Vanden Heuvel et al., 2016*). Using a sample size determined *a priori* to detect the effect based on the originally reported data, the comparison of the effective concentrations of Ewing's sarcoma cell lines to osteosarcoma cell lines was not statistically significant ($U$ = 12, $p$=0.390). Thus, the null hypothesis that the effective concentration of olaparib is similar for Ewing's sarcoma and osteosarcoma cell lines can not be rejected. A similar result was obtained with a duplicate set of plates using the same cell lines (*Figure 1—figure supplement 1*). To summarize, for this experiment we found results that were in the same direction as the original study and not statistically significant.

These results should take into consideration that other methods to assess olaparib sensitivity, such as cellular viability assays, did not result in all Ewing's sarcoma cell lines tested being acutely sensitive to PARP inhibitors, but rather a majority displaying increased sensitivity with some cell lines

relatively insensitive (*Garnett et al., 2012*; *Gill et al., 2015*). Importantly, the cellular viability assays utilized a larger sample size compared to the colony formation assay reported here. The sample size used for this replication attempt was determined from the effect reported in the original study between Ewing's sarcoma and osteosarcoma cell lines. Although the replication study was powered to detect the original effect size estimate with at least 80% power, both the original study and this replication used relatively small sample sizes, which can confound the findings and prevent them from being extrapolated to the overall population (*Button et al., 2013*; *Faber and Fonseca, 2014*). Similarly, only five of the cell lines tested were the same between the two studies, largely due to the inability to obtain any of the Ewing' sarcoma cell lines used in the original study. These factors, among others, influence the research outcome of each designed experiment. Further studies should take into account both of these results, especially when considering the number of cell lines to test.

## Sensitivity to olaparib in cells transformed with the *EWS-FLI1* rearrangement

The *EWS-FLI1* rearrangement is characteristic of Ewing's sarcoma tumors and in the original study was identified as a statistically significant association with olaparib sensitivity (*Garnett et al., 2012*). To test whether the sensitivity to olaparib was due to the *EWS-FLI1* rearrangement, we independently replicated an experiment comparing olaparib sensitivity in mouse mesenchymal cells transformed with *EWS-FLI1*, or *FUS-CHOP* a related liposarcoma-associated translocation (*Riggi et al., 2006*; *2005*). This experiment is similar to what was reported in Figure 4E of *Garnett et al. (2012)* and described in Protocol 2 in the Registered Report (*Vanden Heuvel et al., 2016*). Using the same transformed mouse mesenchymal cells as the original study, as well as the human Ewing's sarcoma cell line SK-N-MC for comparison, sensitivity to olaparib was determined using a cellular viability assay over a range of concentrations. During the course of the assay, the cells continued to proliferate in no drug conditions (*Figure 2—figure supplement 1*), a necessary condition since olaparib sensitivity relies on cellular proliferation (*Dale Rein et al., 2015*; *Weston et al., 2010*). While the SK-N-MC cells displayed sensitivity to olaparib with a mean $IC_{50}$ of 2.67 μM, 95% CI [3.11–2.23], the *EWS-FLI1* and *FUS-CHOP* transformed mouse mesenchymal cells were both relatively resistant with more than 50% of cells remaining viable at the highest dose tested (*Figure 2*). This compares to the original study that reported *EWS-FLI1* transformed mouse mesenchymal cells displaying olaparib sensitivity similar to SK-N-MC cells (estimated $IC_{50}$ of 1.1 μM for *EWS-FLI1* and 1.5 μM for SK-N-MC) with *FUS-CHOP* transformed cells remaining relatively resistant (estimated $IC_{50}$ of 7.8 μM) (*Garnett et al., 2012*). The analysis plan specified in the Registered Report (*Vanden Heuvel et al., 2016*) proposed to compare the $IC_{50}$ values from *EWS-FLI1* transformed cells, or SK-N-MC cells, to *FUS-CHOP* transformed cells, however as stated above this could not be performed because of the inability to determine $IC_{50}$ values for either of the transformed mouse mesenchymal cells. To summarize, for this experiment we found results that were not in the same direction as the original study.

## Olaparib sensitivity after depletion of *EWS-FLI1* from A673 cells

To test if *EWS-FLI1* is necessary for olaparib sensitivity, we replicated an experiment similar to what was reported in Figure 4F and Supplemental Figure 20 in *Garnett et al. (2012)* and described in Protocol 3 in the Registered Report (*Vanden Heuvel et al., 2016*). Using the Ewing's sarcoma cell line, A673, which harbors the *EWS-FLI1* translocation, cells were transfected with either siRNA targeting the *EWS-FLI1* fusion or control siRNA and then concomitantly treated with a range of concentrations of olaparib or vehicle control. Knockdown efficiency was examined by quantitative real-time-polymerase chain reaction (qRT-PCR). A673 cells transfected with *EWS-FLI1* siRNA resulted in an average reduction of 65% in *EWS-FLI1* expression relative to control siRNA whether cells were treated with vehicle control or olaparib (*Figure 3B*). Cellular viability was decreased by olaparib treatment in a dose dependent manner with *EWS-FLI1* and control depleted cells displaying similar sensitivities (*Figure 3A*, *Figure 3—figure supplement 1*). This compares to the original study that reported olaparib sensitivity was partially reverted when *EWS-FLI1* was depleted from A673 cells (estimated $IC_{50}$ of 1.4 μM for control siRNA and 2.7 μM for *EWS-FLI1* siRNA) (*Garnett et al., 2012*). The original study also reported an achieved knockdown of ~94% in *EWS-FLI1* expression whether cells were treated with olaparib or vehicle control (*Garnett et al., 2012*). The difference in achieved knockdown between the original study and this replication attempt is a possible reason for the

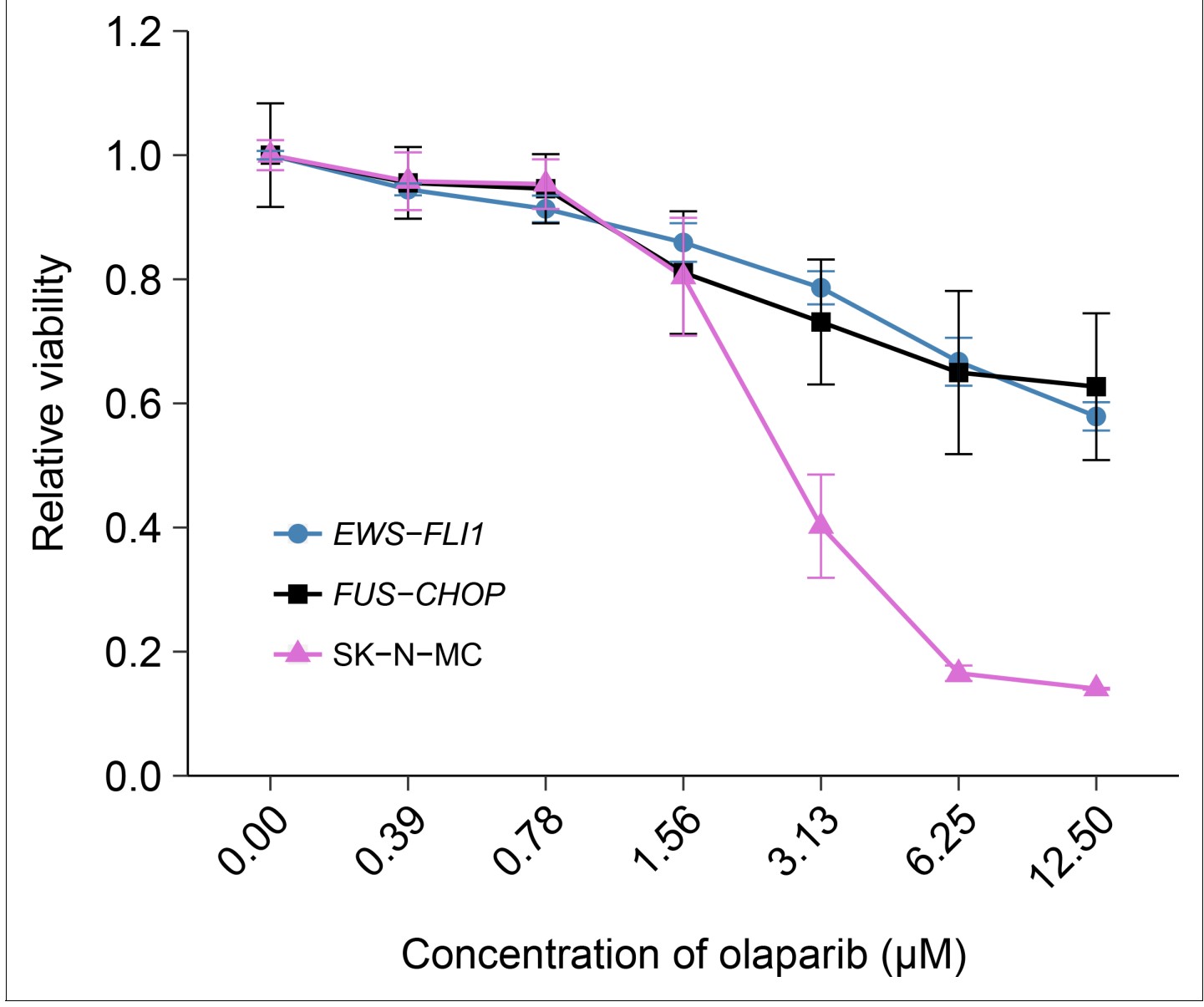

**Figure 2.** Olaparib sensitivity in cells transformed with the *EWS-FLI1* rearrangement. Cell viability assays were performed for *EWS-FLI1* and *FUS-CHOP* transformed mouse mesenchymal cells, as well as the human Ewing's sarcoma cell line (SK-N-MC), which harbors the *EWS-FLI1* fusion. Cells were treated with the indicated doses of olaparib and 72 hr later cell viability was determined. Relative viability was calculated as a percentage of vehicle control treated cells. Means reported and error bars represent *SD* from three independent biological repeats. Additional details for this experiment can be found at https://osf.io/t3dm6/.

DOI: https://doi.org/10.7554/eLife.29747.004

The following figure supplement is available for figure 2:

**Figure supplement 1.** Population doubling time of cells and confirmation of *EWS-FLI1* rearrangement.

DOI: https://doi.org/10.7554/eLife.29747.005

differences in olaparib sensitivity outcomes. The level of knockdown required to yield a given phenotype varies because it is system-dependent (e.g. cell type, assay, function of gene-of-interest), thus, a higher level of knockdown might be required to observe an effect with this experimental design. Importantly, though, observing different outcomes are informative to establish the range of conditions under which a given phenotype can be observed (*Bailoo et al., 2014*).

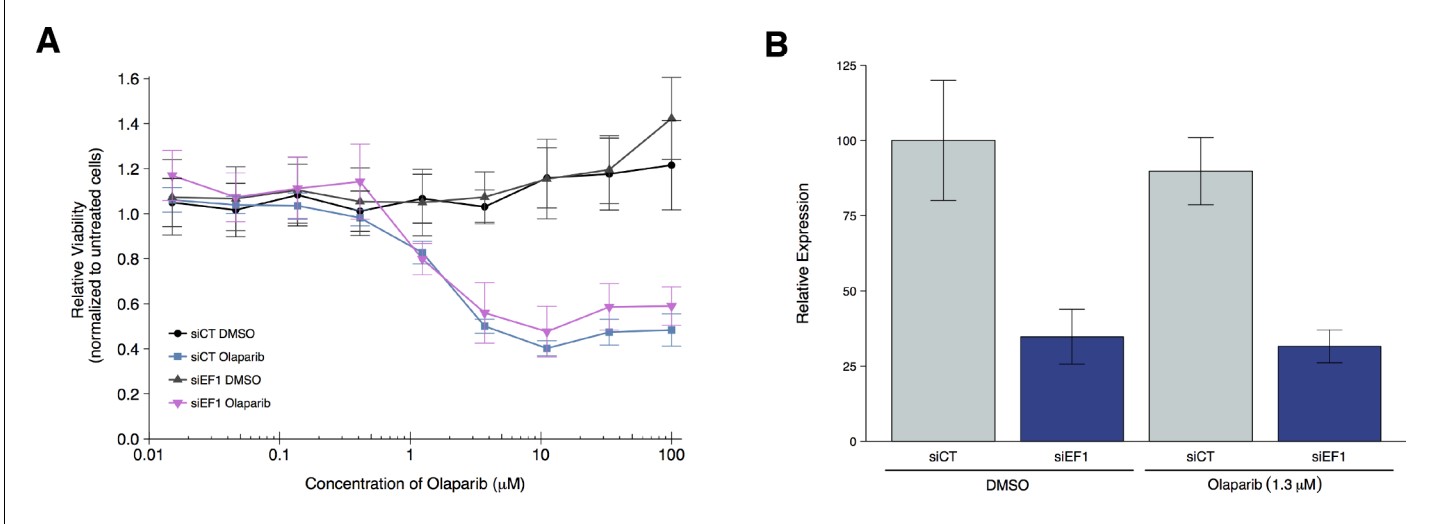

**Figure 3.** Olaparib sensitivity after depletion of *EWS-FLI1* from A673 cells. A673 Ewing's sarcoma cells transiently transfected with negative control siRNA (siCT) or an siRNA targeting the *EWS-FLI1* translocation (siEF1). (**A**) Cell viability assays were performed with the indicated doses of olaparib, or an equivalent volume of vehicle control (DMSO). After 72 hr of treatment cell viability was determined. Relative viability was calculated as a percentage of untreated cells. Means reported and error bars represent *SD* from three independent biological repeats. (**B**) siRNA-mediated depletion of *EWS-FLI1* was determined after 72 hr treatment with vehicle control (DMSO) or olaparib. Relative expression levels of *EWS-FLI1* expression normalized to *RPLP0* (ribosomal protein lateral stalk subunit P0) was determined by qRT-PCR. Expression level of siCT cells treated with DMSO was assigned a value of 100. Means reported and error bars represent *SD* from three independent biological repeats. Two-way ANOVA main effect for siRNA (siCT or siEF1); $F(1,12)$ = 96, $p=4.46\times10^{-7}$. Pairwise contrast between DMSO treated cells transfected with siCT or siEF1; $t(12)$ = 7.32, uncorrected $p=9.17\times10^{-6}$ with *a priori* alpha level = 0.025; (Bonferroni corrected $p=1.83\times10^{-5}$). Pairwise contrast between olaparib treated cells transfected with siCT or siEF1; $t(12)$ = 6.53, uncorrected $p=2.80\times10^{-5}$ with *a priori* alpha level = 0.025; (Bonferroni corrected $p=5.60\times10^{-5}$). Additional details for this experiment can be found at https://osf.io/2w22x/.

DOI: https://doi.org/10.7554/eLife.29747.006

The following figure supplement is available for figure 3:

**Figure supplement 1.** Cell viability assays for each biological repeat.

DOI: https://doi.org/10.7554/eLife.29747.007

It's also worth noting that in the original study, olaparib treatment in the control siRNA cells resulted in a relative viability of zero at the highest concentrations of olaparib tested, while in this replication attempt relative viability remained at around 40% (*Figure 3A*). This could be due to how the lower bound of detection (maximal response) was determined (*Sebaugh, 2011*) as well as differences in cellular growth conditions that could impact olaparib sensitivity (*Stordal et al., 2013*; *Strese et al., 2013*). The analysis plan specified in the Registered Report (*Vanden Heuvel et al., 2016*) proposed to compare the $IC_{50}$ values from control siRNA transfected cells to *EWS-FLI1* siRNA transfected cells, however this could not be performed because of the inability to determine absolute $IC_{50}$ values for either condition following published guidelines (*Sebaugh, 2011*). To summarize, for this experiment we found the cell viability results were not in the same direction as the original study.

## Meta-analysis of original and replication effects

We performed a meta-analysis using a random-effects model, where possible, to combine each of the effects described above as pre-specified in the confirmatory analysis plan (*Vanden Heuvel et al., 2016*). To provide a standardized measure of the effect, a common effect size was calculated for each effect from the original and replication studies. Cliff's delta (*d*) is a non-parametric estimate of effect size that measures how often a value in one group is larger than the values from another group. The estimate of the effect size of one study, as well as the associated uncertainty (i.e. confidence interval), compared to the effect size of the other study provides another approach to compare the original and replication results (*Errington et al., 2014*; *Valentine et al., 2011*). Importantly,

the width of the confidence interval for each study is a reflection of not only the confidence level (e.g. 95%), but also variability of the sample (e.g. *SD*) and sample size.

The comparison of the effective concentration of olaparib for Ewing's sarcoma cells to osteosarcoma cells resulted in a Cliff's *d* = 0.34, 95% CI [−0.24, 0.74] for this study, whereas Cliff's *d* = 0.93, 95% CI [0.62, 0.99] for the data estimated *a priori* from Figure 4C of the original study (*Garnett et al., 2012*). A meta-analysis (*Figure 4*) of these two effects resulted in Cliff's *d* = 0.63, 95% CI [−0.10, 0.92], which was statistically significant (*p*=0.029). Importantly, the confidence interval around Cliff's *d* is asymmetric, while the *p* value is calculated using the normal distribution and is thus not well defined; however there is no agreement on how to compute *p* values from an asymmetric distribution (*Dunne et al., 1996*; *Rohatgi and Saleh, 2000*). Both results are consistent when considering the direction of the effect, however the point estimate of the replication effect size was not within the confidence interval of the original result, or vice versa.

This direct replication provides an opportunity to understand the present evidence of these effects. Any known differences, including reagents and protocol differences, were identified prior to conducting the experimental work and described in the Registered Report (*Vanden Heuvel et al., 2016*). However, this is limited to what was obtainable from the original paper and through communication with the original authors, which means there might be particular features of the original experimental protocol that could be critical, but unidentified. So while some aspects, such as cell lines, number of cells plated/injected, and the specific PARP inhibitor were maintained, others were unknown or not easily controlled for. These include variables such as cell line genetic drift (*Hughes et al., 2007*; *Kleensang et al., 2016*), sex of cell lines (*Clayton and Collins, 2014*), impacts of atmospheric oxygen on cell viability and growth (*Boregowda et al., 2012*), and differing

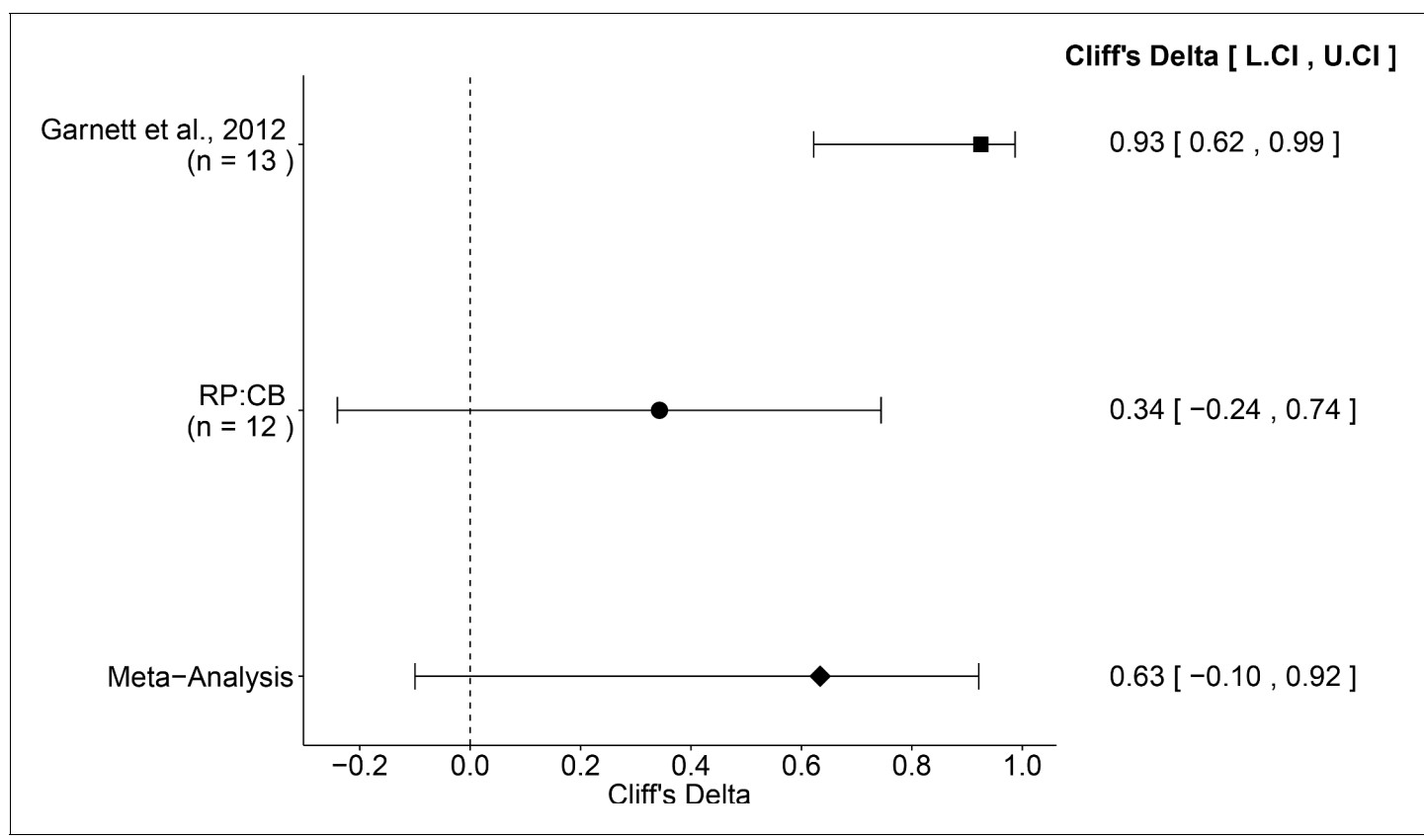

**Figure 4.** Meta-analysis of effect. Effect size and 95% confidence interval are presented for *Garnett et al. (2012)*, this replication attempt (RP:CB), and a random effects meta-analysis of those two effects. Sample sizes used in *Garnett et al. (2012)* and this replication attempt are reported under the study name. Random effects meta-analysis of effective concentrations of Ewing's sarcoma cell lines to osteosarcoma cell lines from colony formation assays (meta-analysis *p*=0.029). Additional details for this meta-analysis can be found at https://osf.io/whs6e/.
DOI: https://doi.org/10.7554/eLife.29747.008

compound potency resulting from different stock solutions (*Kannt and Wieland, 2016*) or from variation in cell division rates (*Hafner et al., 2016*). Whether these or other factors influence the outcomes of this study is open to hypothesizing and further investigation, which is facilitated by direct replications and transparent reporting.

## Materials and methods

As described in the Registered Report (*Vanden Heuvel et al., 2016*), we attempted a replication of the experiments reported in Figure 4C and E–F, and Supplemental Figures 16 and 20 of *Garnett et al. (2012)*. A detailed description of all protocols can be found in the Registered Report (*Vanden Heuvel et al., 2016*). Additional detailed experimental notes, data, and analysis are available on the Open Science Framework (OSF) (RRID:SCR_003238) (https://osf.io/nbryi/; *Vanden Heuvel et al., 2017*).

### Cell culture

A673 (ATCC, cat# CRL-1598, RRID:CVCL_0080), TC-71 (Children's Oncology Group Cell Culture and Xenograft Repository (COGcell), RRID:CVCL_2213) (*Whang-Peng et al., 1986*), SK-N-MC (ATCC, cat# HTB-10, RRID:CVCL_0530), CHLA-9 (COGcell, RRID:CVCL_M150) (*Batra et al., 2004*), CHLA-10 (COGcell, RRID:CVCL_6583) (*Batra et al., 2004*), U-2-OS (ATCC, cat# HTB-96, RRID:CVCL_0042), SJSA-1 (ATCC, cat# CRL-2098, RRID:CVCL_1697), SAOS-2 (ATCC, cat# HTB-85, RRID:CVCL_0548), HOS (ATCC, cat# CRL-1543, RRID:CVCL_0312), MG-63 (ATCC, cat# CRL-1427, RRID:CVCL_0426), 143B (ATCC, cat# CRL-8303, RRID:CVCL_2270), G-292 clone A141B1 (ATCC, cat# CRL-1423, RRID:CVCL_2909), DoTc2-4510 (ATCC, cat# CRL-7920, RRID:CVCL_1181), MES-SA (ATCC, cat# CRL-1976, RRID:CVCL_1404), and *EWS-FLI1* and *FUS-CHOP* transformed mouse mesenchymal progenitor cells (shared by Stamenkovic lab, Centre Hospitalier Universitaire Vaudois (CHUV) from vials frozen in 2006) were maintained in growth medium as described in the Registered Report (*Vanden Heuvel et al., 2016*) with Fetal Bovine Serum (FBS) (Gibco, cat# 16000–036), Roswell Park Memorial Institute (RPMI) 1640 medium (Gibco, cat# A10491), and Dulbecco's Modified Eagle's Medium (DMEM), high-glucose (Gibco, cat# 11965–092) sourced differently than listed. A673 (*Giard et al., 1973*) and SK-N-MC cells (*Biedler et al., 1973*) were originally classified as neuroblastoma cell lines, but have since been determined to be Ewing's sarcoma cells (*Martínez-Ramírez et al., 2003*; *Whang-Peng et al., 1986*). Cells were grown at 37°C in a humidified atmosphere at 5% $CO_2$. Quality control data for cell lines are available at https://osf.io/x4zwb/. This includes results confirming the cell lines were free of mycoplasma contamination and common mouse pathogens (IDEXX BioResearch, Columbia, Missouri). Additionally, STR DNA profiling of the cell lines was performed and all cells were confirmed to be the indicated cell lines when queried against STR profile databases.

### Therapeutic compounds

10 mM olaparib (Selleckchem, cat# S1060; lot# S106021) was aliquoted into amber vials with O-ring screw caps, sparged with Argon, and stored at −80°C until use. Freeze-thaws were limited to less than five times.

### Colony formation assays

Cells were plated at low density (2000 cells per well of 6-well culture plate (Falcon, cat# 353224)) and allowed to adhere overnight. The following day cells were treated with five doses of olaparib (10 µM, 2 µM, 1 µM, 0.32 µM, 0.1 µM) diluted to give a final DMSO concentration of 0.1% v/v or vehicle control (0.1% v/v DMSO). Medium was replaced and cells were retreated with olaparib or vehicle control every 3 or 4 days. When sufficient colonies were visible (greater than ~100) in the vehicle control condition (after 7–21 days), cells were washed with PBS and fixed with ice-cold methanol for 30 min while shaking at room temperature. Methanol was removed and cells were stained with Giemsa stain at 1:20 dilution in deionized water for 4 hr at room temperature (or overnight at 4°C) while shaking. Cells were rinsed with water, dried, and colonies were quantified (ImageJ software (RRID:SCR_003070), version 1.51h (*Schneider et al., 2012*)) in a blinded manner for each well. The effective concentration for each cell line was determined as the concentration that reduced colony formation by greater than 90% compared to vehicle control. For G-292 cells, where the highest

olaparib concentration tested (10 µM) did not inhibit colony formation by at least 90%, the effective concentration was determined as 10 µM. This assay was performed with a duplicate set of plates using the same cell lines. Images of stained plates with cell line and olaparib concentration labeled, similar to Supplemental Figures 16 of *Garnett et al. (2012)*, are available at https://osf.io/whwsk/.

## Cell viability assays with mouse mesenchymal progenitor cells

The seeding density of each cell line was empirically determined by seeding between 500 and $1.6 \times 10^4$ cells in 96 well plates (Costar, cat# 3917) in 100 µl medium in technical quadruplicate. 48, 72, and 96 hr after seeding, cells were fixed in 4% paraformaldehyde (Affymetrix, cat# 19943) for 30 min at 37°C. Cells were stained with 1 µM Syto 60 red fluorescent nucleic acid stain, diluted in PBS, for 1 hr according to manufacturer's instructions. Fluorescence signal was quantified using a red fluorescent filter (excitation = 625 nm/emission = 660–720 nm) and a GloMAX Multi Detection System (Promega, model# 9311–011), software version 2.3.2. For each cell line, a log(dose) response curve was fitted using the 'log(Agonist) vs. response – Find ECanything' analysis from GraphPad Prism software (San Diego, California, RRID:SCR_002798), version 6.0b. The number of cells seeded that achieved a 70% fluorescent signal (~70% confluent) after 96 hr was determined.

SK-N-MC cells were seeded at 8000 or 12,000 cells/well and *EWS-FLI1* and *FUS-CHOP* transformed mouse mesenchymal progenitor cells were seeded at 3000 or 4,000 cells/well (non-edge wells) into a 96-well plate and incubated overnight. The following day cells were treated with serial dilutions of olaparib to yield six dilutions ranging from 0.39 µM to 12.5 µM to give a final DMSO concentration of 0.1% v/v, as well as vehicle control (0.1% v/v DMSO) and untreated cells that were used to normalize each biological repeat. All conditions were done in technical triplicate. Wells were fixed, stained, and fluorescence signal quantified as described for the seeding density step 24, 48, and 72 hr after start of treatment. Untreated and vehicle control treated wells were used to determine the population doubling time (*Figure 2—figure supplement 1*) using the formula (doubling time = incubation time*ln(2)/ln(72 hr plate average reading/24 hr plate average reading)). Relative viability was calculated as a percentage of vehicle control treated cells. Spline interpolation was performed on values for each biological repeat, where possible, to determine an estimate $IC_{50}$ as described in the Registered Report (*Vanden Heuvel et al., 2016*) with R software (RRID:SCR_001905), version 3.3.2 (*R Core Team, 2017*).

## siRNA transfection and cell viability assays with A673 cells

The A673 Ewing's sarcoma cell line was seeded at 5,000 cells per well in a 96 well plate (non-edge wells) and immediately reverse transfected with 25 nM AllStars negative control siRNA (Qiagen, cat# 1027281) or an siRNA targeting the *EWS-FLI1* translocation (Qiagen, cat# 1027423; custom order: 5'-GGCAGCAGAACCCUUCUUACG-3') with Lipofectamine RNAiMAX with the cells in suspension according to manufacturer's instructions. Immediately after siRNA transfection cells were treated with serial dilutions of olaparib to yield nine dilutions ranging from 100 µM to 0.015 µM or an equivalent volume of vehicle control (DMSO). Untreated cells as well as medium alone wells that were used for background subtraction were included. All conditions were done in technical triplicate. After 72 hr, cell viability was determined using Cell Titer 96 Aqueous One Solution Cell Proliferation Assay (Promega, cat# G3582) according to manufacturer's instructions. After 4 hr incubation at 37°C in a humidified atmosphere at 5% $CO_2$ absorbance was read at 490 nm using a GloMax Multi Detection System. Relative viability was calculated as a percentage of untreated cells after background subtraction (medium only wells).

## Gene expression analysis

In parallel to the cell viability assay, A673 cells were seeded at $3 \times 10^4$ cells per well of a 24 well plate (Falcon, cat# 353226) and immediately reverse transfected with negative control siRNA or an siRNA targeting the *EWS-FLI1* translocation as described above. Immediately after siRNA transfection cells were treated with 1.3 µM olaparib or an equivalent volume of vehicle control (0.013% v/v DMSO). After 72 hr, cells were harvested and RNA isolated using the NucleoSpin RNAII kit (Machery-Nagel, cat# 740955.5) according to manufacturer's instructions. RNA concentration and purity was determined (data available at https://osf.io/ryvmu/). Total RNA was reverse transcribed into cDNA using High-capacity cDNA reverse transcription kit (Applied Biosystems, cat# 4368814)

according to manufacturer's instructions. qRT-PCR reactions were then performed in technical triplicate using *EWS-FLI1* and *RPLP0*-specific primers (Integrated DNA Technologies: sequences listed in Registered Report [*Vanden Heuvel et al., 2016*]), and Perfecta SYBR green FastMix (Quantra Biosciences, cat# 95073–012) according to manufacturer's instructions. PCR cycling conditions were used as follows: [1 cycle 95°C for 10 min – 40 cycles 95°C for 15 s, 60°C 60 s] using a StepOnePlus real-time PCR system (Applied Biosystems, cat# 4376592) and StepOne software (RRID:SCR_014281), version 2.3. *EWS-FLI1* transcript levels were normalized to *RPLP0* levels in each sample.

## Confirmation of *EWS-FLI1* rearrangement in mouse mesenchymal progenitor cells

Presence of the *EWS-FLI1* rearrangement in the mouse mesenchymal progenitor cells shared for this replication attempt was confirmed by qPCR analysis (*Figure 2—figure supplement 1B*). SK-N-MC cells and *EWS-FLI1* and *FUS-CHOP* transformed mouse mesenchymal progenitor cells ($>3 \times 10^6$ cells) were pelleted at 110x*g* for 5 min, washed once with PBS, and lysed in 400 µl RNA lysis buffer (Promega, cat# Z3051). RNA extraction was performed with the SV96 Total RNA Isolation System (Promega, cat# Z3505) according to manufacturer's instructions. RNA preps were eluted in 100 µl nuclease free water (Promega, cat# P119E). RNA concentration and purity was determined with a NanoDrop 2000 (Thermo Fisher Scientific, Waltham, Massachusetts) (data available at https://osf.io/2kz7n/). Total RNA was reverse transcribed into cDNA using High-capacity cDNA reverse transcription kit according to manufacturer's instructions. qRT-PCR were then performed in technical triplicate using *EWS-FLI1* and mouse *Actb* specific primers (*EWS-FLI1* primer is same as above; *Actb* sequence is: Forward: 5'-GACTCATCGTACTCCTGCTTG-3', Reverse: 5'-GATTACTGCTCTGGCTCCTAG-3') at 200 nM final concentration, and Perfecta SYBR green FastMix according to manufacturer's instructions. PCR conditions were the same as described above. Negative controls containing water with no cDNA template, as well as cDNA template without reverse transcriptase were included.

## Statistical analysis

Statistical analysis was performed with R software (RRID:SCR_001905), version 3.3.2 (*R Core Team, 2017*). All data, csv files, and analysis scripts are available on the OSF (https://osf.io/nbryi/). Confirmatory statistical analysis was pre-registered (https://osf.io/wt8df/) before the experimental work began as outlined in the Registered Report (*Vanden Heuvel et al., 2016*). Data were checked to ensure assumptions of statistical tests were met. When described in the results, the Bonferroni correction, to account for multiple testings, was applied to the alpha error or the *p*-value. The Bonferroni corrected value was determined by divided the uncorrected value (.05) by the number of tests performed. A meta-analysis of a common original and replication effect size was performed with a random effects model and the *metafor* R package (*Viechtbauer, 2010*) (https://osf.io/whs6e/). For the colony formation analysis, to obtain an overall effect size estimate from this replication attempt, the estimates from the two repeats were combined with a fixed effects model. Furthermore, all meta-analyses were performed without weighting, since unweighted Cliff's *d* has been reported to reduce bias (*Kromrey et al., 2005*). The asymmetric confidence intervals for the overall Cliff's *d* estimate was determined using the normal deviate corresponding to the (1 - alpha/2)th percentile of the normal distribution (*Cliff, 1993*). The original study data presented in Figure 4C was extracted *a priori* from the published figure by determining the height of each bar, while the summary data (mean and standard deviation) pertaining to Figure 4E-F were shared by the original authors. The data and estimated $IC_{50}$ values for Figure 4E-F were published in the Registered Report (*Vanden Heuvel et al., 2016*) and used in the power calculations to determine the sample size for this study.

## Deviations from registered report

The source of FBS, RPMI 1640, DMEM, high-glucose, and SYBR Green PCR mix were different than what is listed in the Registered Report, with the used source and catalog number listed above. Statistical analysis proposed in the Registered Report for some of the experiments were unable to be performed as described above. We also included an additional test to confirm the *EWS-FLI1* rearrangement in the mouse mesenchymal progenitor cells shared for this replication attempt.

Additional materials and instrumentation not listed in the Registered Report, but needed during experimentation are also listed.

## Acknowledgements

The Reproducibility Project: Cancer Biology would like to thank Dr. Cyril Benes (Massachusetts General Hospital Cancer Center) for sharing critical information and data and Dr. Ivan Stamenkovic (CHUV) for sharing the transformed mouse mesenchymal progenitor cells. We would also like to thank the following companies for generously donating reagents to the Reproducibility Project: Cancer Biology; American Type and Tissue Collection (ATCC), Applied Biological Materials, BioLegend, Charles River Laboratories, Corning Incorporated, DDC Medical, EMD Millipore, Harlan Laboratories, LI-COR Biosciences, Mirus Bio, Novus Biologicals, Sigma-Aldrich, and System Biosciences (SBI).

## Additional information

### Group author details

**Reproducibility Project: Cancer Biology**
Elizabeth Iorns: Science Exchange, Palo Alto, United States; Rachel Tsui: Science Exchange, Palo Alto, United States; Alexandria Denis: Center for Open Science, Charlottesville, United States; Nicole Perfito: Science Exchange, Palo Alto, United States; Timothy M Errington: Center for Open Science, Charlottesville, United States

### Competing interests

John P Vanden Heuvel, Ewa Maddox, Samar W Maalouf: Indigo Biosciences is a Science Exchange associated lab. Reproducibility Project: Cancer Biology: EI, RT, NP: Employed by and hold shares in Science Exchange Inc.The other authors declare that no competing interests exist.

### Funding

| Funder | Author |
| --- | --- |
| Laura and John Arnold Foundation | Reproducibility Project: Cancer Biology |

The funder had no role in study design, data collection and interpretation, or the decision to submit the work for publication.

### Author contributions

John P Vanden Heuvel, Acquisition of data, Drafting or revising the article; Ewa Maddox, Samar W Maalouf, Acquisition of data; Reproducibility Project: Cancer Biology, Analysis and interpretation of data, Drafting or revising the article

### Author ORCIDs

Alexandria Denis http://orcid.org/0000-0002-1210-2309
Timothy M Errington http://orcid.org/0000-0002-4959-5143

### Decision letter and Author response

Decision letter https://doi.org/10.7554/eLife.29747.012
Author response https://doi.org/10.7554/eLife.29747.013

## Additional files

### Supplementary files

• Transparent reporting form
DOI: https://doi.org/10.7554/eLife.29747.009

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
