## [Decision Letter]

Thank you for submitting your article "Replication Study: Systematic identification of genomic markers of drug sensitivity in cancer cells" for consideration by *eLife*. Your article has been reviewed by three peer reviewers, and the evaluation has been overseen by a Reviewing Editor and Charles Sawyers as the Senior Editor. The following individuals involved in review of your submission have agreed to reveal their identity: Jenny Barrett (Reviewer #1); Cyril Benes (Reviewer #2).

The reviewers have discussed the reviews with one another and the Reviewing Editor has drafted this decision to help you prepare a revised submission.

Summary of discussion:

This replication study reports the results of a series of experiments aimed at reproducing findings in an original manuscript by Garnett et al. (Nature 2012) reporting increased sensitivity to PARP inhibitors among Ewing's sarcoma cell lines.

After thorough discussion among reviewers and members of the Editorial board, it was decided to request submission of a revised manuscript that explicitly states the inconclusive nature of the replication study.

The replication study reports increased sensitivity for PARP inhibitors among select Ewing's cell lines, but the difference is not statistically significant as reported in the original publication, which could certainly be attributed to the use of different and fewer sarcoma cell lines. Another key experiment involving knockdown of EWS-FLI is also inconclusive, due to the relatively poor siRNA knockdown of EWS-FLI relative to the original paper.

It is then required that these differences in the experimental approach be made very clear in the revised manuscript, and that the word "inconclusive" be part of the Abstract since the report does not disprove the hypothesis that Ewing's sarcoma cell lines are more sensitive to PARP inhibitors.

Although additional experiments cannot be requested at this point, the revised Discussion should include a statement saying that more cell lines would need to be tested to move beyond the current "inconclusive" nature of the report, as well as a more robust EWS-FLI knockdown.

[Editors' note: further revisions were requested prior to acceptance, as described below.]

Thank you for resubmitting your work entitled "Replication Study: Systematic identification of genomic markers of drug sensitivity in cancer cells" for further consideration at *eLife*. Your revised article has been favorably evaluated by Charles Sawyers (Senior Editor) and a Reviewing editor.

The manuscript has been improved and is essentially ready for acceptance provided you make editorial changes to the Abstract acknowledging that some of the results, particularly the siRNA experiments are inconclusive due to difference in knockdown efficiency. It is important that this be explicitly stated to avoid any misinterpretation of the conclusions. (The current concluding sentence in the Abstract, "No single study can provide conclusive evidence for or against a claim…" does not provide the necessary level of clarity.)

---

## [Author Response]

Summary of discussion:This replication study reports the results of a series of experiments aimed at reproducing findings in an original manuscript by Garnett et al. (Nature 2012) reporting increased sensitivity to PARP inhibitors among Ewing's sarcoma cell lines.After thorough discussion among reviewers and members of the Editorial board, it was decided to request submission of a revised manuscript that explicitly states the inconclusive nature of the replication study.The replication study reports increased sensitivity for PARP inhibitors among select Ewing's cell lines, but the difference is not statistically significant as reported in the original publication, which could certainly be attributed to the use of different and fewer sarcoma cell lines. Another key experiment involving knockdown of EWS-FLI is also inconclusive, due to the relatively poor siRNA knockdown of EWS-FLI relative to the original paper.It is then required that these differences in the experimental approach be made very clear in the revised manuscript, and that the word "inconclusive" be part of the Abstract since the report does not disprove the hypothesis that Ewing's sarcoma cell lines are more sensitive to PARP inhibitors.Although additional experiments cannot be requested at this point, the revised Discussion should include a statement saying that more cell lines would need to be tested to move beyond the current "inconclusive" nature of the report, as well as a more robust EWS-FLI knockdown.

We have revised the manuscript to include further discussion on the sample size and selection of cell lines used in this replication and the original study for the colony survival assay and how this can influence a result. Similarly we revised the siRNA knockdown of EWS-FLI1 to include discussion about what conditions are necessary to observe an outcome, specifically how the knockdown percentage might influence the ability to detect an effect. These experiments are interpretable and the comments the reviewers raised are some of the possible explanations for the difference in results between the two studies and what the necessary conditions are to observe the effect reported in the original study.

We have also revised the Abstract and final paragraph of the Discussion to highlight how this study, just like any single study, is not conclusive evidence for or against an effect, but that the accumulation of evidence provides the support for or against a claim and the conditions necessary to observe it.

[Editors' note: further revisions were requested prior to acceptance, as described below.]

The manuscript has been improved and is essentially ready for acceptance provided you make editorial changes to the Abstract acknowledging that some of the results, particularly the siRNA experiments are inconclusive due to difference in knockdown efficiency. It is important that this be explicitly stated to avoid any misinterpretation of the conclusions. (The current concluding sentence in the Abstract, "No single study can provide conclusive evidence for or against a claim…" does not provide the necessary level of clarity.)

We disagree with stating that results are 'inconclusive', regardless of what study or experiment it is in reference to, particularly if it is in the context of equating 'statistically significant' and 'positive results' with 'conclusive' and 'null' with 'inconclusive'. For example, failing to reject the null hypothesis (that Ewing's Sarcoma and other tested cell lines are equivalently sensitive to olaparib) does not mean we accept the null hypothesis, only that there is insufficient evidence at the predefined level of significance and power (sample size) to reject the null hypothesis. We do agree that the results we report could be attributed to the use of different sarcoma cell lines and a level of siRNA knockdown of EWS-FLI relative to the original study and have revised the Abstract to highlight this.